

# Evaluation of *in vitro* osteoblast and osteoclast differentiation from stem cell: a systematic review of morphological assays and staining techniques

Shahrul Hisham Zainal Ariffin[1,*], Rohaya Megat Abdul Wahab[2,*], Muhammad Abdul Razak[3], Muhammad Dain Yazid[4], Muhammad Ashraf Shahidan[1], Azizi Miskon[5] and Intan Zarina Zainol Abidin[6]

[1] Department of Science Biology and Biotechnology, Faculty of Science and Technology, Universiti Kebangsaan Malaysia, Bangi, Selangor, Malaysia
[2] Centre of Family Dental Health, Faculty of Dentistry, Universiti Kebangsaan Malaysia, Kuala Lumpur, Wilayah Persekutuan Kuala Lumpur, Malaysia
[3] Board of Director Office, 6th Floor, Chancellery Building, Universiti Kebangsaan Malaysia, Bangi, Selangor, Malaysia
[4] Centre for Tissue Engineering and Regenerative Medicine, Universiti Kebangsaan Malaysia Medical Centre, Universiti Kebangsaan Malaysia, Kuala Lumpur, Wilayah Persekutuan Kuala Lumpur, Malaysia
[5] Department of Electrical and Electronics Engineering, Faculty of Engineering, National Defence University of Malaysia, Sungai Besi, Wilayah Persekutuan Kuala Lumpur, Malaysia
[6] Department of Pharmaceutical Sciences, Faculty of Pharmacy, University of Cyberjaya, Cyberjaya, Selangor, Malaysia
* These authors contributed equally to this work.

Corresponding authors
Shahrul Hisham Zainal Ariffin, shahroy8@gmail.com
Intan Zarina Zainol Abidin, izzarina7@gmail.com

## ABSTRACT

**Background.** Understanding human stem cell differentiation into osteoblasts and osteoclasts is crucial for bone regeneration and disease modeling. Numerous morphological techniques have been employed to assess this differentiation, but a comprehensive review of their application and effectiveness is lacking.

**Methods.** Guided by the PRISMA framework, we conducted a rigorous search through the PubMed, Web of Science and Scopus databases, analyzing 254 articles. Each article was scrutinized against pre-defined inclusion criteria, yielding a refined selection of 14 studies worthy of in-depth analysis.

**Results.** The trends in using morphological approaches were identified for analyzing osteoblast and osteoclast differentiation. The three most used techniques for osteoblasts were Alizarin Red S (mineralization; six articles), von Kossa (mineralization; three articles) and alkaline phosphatase (ALP; two articles) followed by one article on Giemsa staining (cell morphology) and finally immunochemistry (three articles involved Vinculin, F-actin and Col1 biomarkers). For osteoclasts, tartrate-resistant acid phosphatase (TRAP staining) has the highest number of articles (six articles), followed by two articles on DAPI staining (cell morphology), and immunochemistry (two articles with VNR, Cathepsin K and TROP2. The study involved four stem cell types: peripheral blood monocyte, mesenchymal, dental pulp, and periodontal ligament.

**Conclusion.** This review offers a valuable resource for researchers, with Alizarin Red S and TRAP staining being the most utilized morphological procedures for osteoblasts and osteoclasts, respectively. This understanding provides a foundation for future research in this rapidly changing field.

## INTRODUCTION

Stem cells (SCs) are undifferentiated cells that can self-renew and differentiate into multi-lineages (*Zainal Ariffin et al., 2017*). They have been widely studied for many years because of their potential to revolutionize the future of the medical field by providing new innovative stem cell therapies (*Yazid et al., 2010*). In-depth stem cell research raises many new scientific questions due to advances in this field that have led to various discoveries. Stem cells can be divided into two large groups: embryonic stem cells and adult stem cells (*Ab Kadir et al., 2012*). These two groups of stem cells share the ability to self-renew and differentiate into specialized cells but have different characteristics from different points of view.

Embryonic stem cells are pluripotent, meaning they can differentiate into any type of cell, and are derived from the inner cell mass during the early stages of the blastocyst development (*Ab Kadir et al., 2011*). Embryonic stem cells can be cultured easily and exhibit unique properties, including spontaneous differentiation into three layers of in-vitro or in-vivo teratoma formation (*Zakrzewski et al., 2019*). In contrast, adult stem cells are multipotent stem cells with certain limitations in differentiation. The primary role of adult stem cells is to maintain and repair damaged tissue. The ability of adult stem cells to differentiate is limited; these cells can be multipotent or unipotent (*Prochazkova et al., 2015*). Presently, adult stem cells are the focus of most stem cell research because adult stem cells offer hope for stem cell therapy to treat disease in the future and ethical issues do not preclude its use. Adult stem cells can be obtained from a variety of sources such as peripheral blood (PB) (*Zainol Abidin et al., 2010*; *Yusof et al., 2011*; *Muniandy et al., 2013*; *Zainal Ariffin et al., 2016*), dental pulp tissue (*Nur Akmal et al., 2014*; *Kermani et al., 2014*; *Hagar et al., 2021*; *Intan Zarina et al., 2022*), bone marrow (BM), and periodontal ligament tissue (*Chong et al., 2012*).

The bone is an intricate, multipurpose organ that stores minerals, protects internal organs, and offers mechanical support for movement (*Ansari, Ito & Hofmann, 2021*). While many different kinds of cells and substances play a role in the process of regenerating bone, the two primary cell types involved are osteoblasts and osteoclasts. Osteoblasts are in charge of creating new bone, whereas osteoclasts are in charge of resorbing old bone (*Matsuoka et al., 2014*). However, an imbalance of this tightly coupled process or any disruptions of this homeostasis caused by the defects in the cell or the interruption of intercellular communications between cells can cause a bone to function abnormally which may lead to bone diseases such as osteoporosis, osteopetrosis, osteogenesis imperfecta and Paget's disease (*Phan, Xu & Zheng, 2004*).

Osteoblast secretes bone organic matrix in the organized epithelial structure and tightly controls the matrix mineralization (*Blair et al., 2016*). As the osteoblast develops, it either chooses to differentiate into bone lining cells which play an important role in localized bone remodeling or further differentiate and develop into osteocytes that are incorporated into

the matrix as living cells (*Creecy, Damrath & Wallace, 2021*). The effective regeneration of bone tissue involves great coordination between the actions of osteoblasts and osteoclasts. Many studies have highlighted the key role of osteoblasts and osteoclasts coupling during the process of bone formation and resorption known as bone remodeling (*Borciani et al., 2020*). Despite their lack of direct contact, osteoclasts and osteoblasts cooperate in a coordinated spatiotemporal way to accomplish the processes of bone remodeling and bone regeneration. The physical link that exists between resorption and the production of new bone during the reversal phase provides evidence in support of this notion (*Borciani et al., 2022*).

The mechanisms that regulate communication between osteoclasts and osteoblasts are critical to the biology of bone cells. The differentiation of pre-osteoblasts into mature osteoblasts can be observed by the expression of bone markers such as alkaline phosphatase (ALP), osteocalcin (OCN), osteopontin (OPN), and collagen type I (*Col1*) (*Katagiri & Takahashi, 2002*; *Zainal Ariffin et al., 2017*). This differentiation process is induced by the activation of transcription factors RUNX2 and osterix. On the other hand, osteoclasts can be distinguished with the help of markers and receptors such as tartrate resistant acid phosphatase (TRAP), calcitonin, vitronectin, cathepsin K, DC-STAMP, and H-ATPase. Some other osteoclast transcription factors include PU-1, cFos, MITF, and NFATc1 (*Phan, Xu & Zheng, 2004*; *Florencio-Silva et al., 2015*). Osteocytes act as biosensors and detect mechanical stress in bones which then secrete bone-resorbing molecule RANKL or osteoprotegerin (OPG), a decoy receptor for RANKL. The binding of RANKL to its receptor RANK triggers osteoclast cells to induce bone resorption through secretion of cathepsin K. To avoid excessive bone resorption, the OPG molecule binds with RANKL to prevent it from interacting with RANK thus reducing osteoclast.

Histology techniques that emphasize morphological and functional characteristics can also be used to observe osteoblastic and osteoclastic differentiation using multiple staining techniques. Although the visualization of shape and function provided by histological techniques provides important insights into the differentiation of osteoblasts and osteoclasts, a noticeable inconsistency exists in the specific analysis methods used across studies. This review aims to address this gap by performing an extensive analysis of the 10 years of research. We meticulously assess and condense specific research focused on human osteoblast and osteoclast differentiation methods. By identifying the most commonly used techniques, we hope to inform future research direction, promoting for more consistent approach to analyzing these important cell types.

## MATERIAL & METHOD

This systematic review, which comprised studies published between 2013 and 2023, was carried out following the Preferred Reporting Items for Systematic Reviews and Meta-Analyses (PRISMA) requirements (*Page et al., 2021*). The following is how the PICOS question was created: Which are the common trends of techniques for morphological characterization of human stem cell differentiation into osteoblasts and osteoclasts among the various methods involved in osteoblast and osteoclast morphology utilizing obtainable

human stem cells? S.H.Z.A. and I.Z.Z.A., two independent researchers, conducted the searches and assessed the articles to ascertain their suitability. Five more authors (M.D.Y., R.M.A.W., A.M., M.A.R. and M.A.S.) assisted in resolving a disagreement with the methodology mentioned.

### Data search

The PubMed, Web of Science (WOS), and Scopus databases contributed to the research articles were included in this systematic review. Utilizing the search engines of these databases, a query string comprised of individual keywords and their combinations was utilized. Table 1 displays the keywords and their combinations that are routinely employed to produce relevant articles.

### Selection requirements and data extraction

The inclusion and exclusion criteria were met with consistency. Review and duplicate articles from both databases were eliminated from this review, retaining only the original publications published in the English language between 2013 and 2023. *In-vitro* studies on the potential of human stem cells to develop into osteoblasts and osteoclasts were included in this review; research utilizing animal cell lines was also excluded. This review did not consider any *in-vivo* research. Studies combining elements of human and animal research were included, however, only the human studies part was included. In addition, PRISMA criteria were followed during the data extraction phase (*Page et al., 2021*).

### Screening process

Duplication of content from other journals was eliminated. To separate any reviews and articles published in languages other than English, the following screening was carried out. Articles that did not follow the guidelines for osteoblastic and osteoclastic differentiation, as well as those that did not make use of human stem cells, were then eliminated. The eligibility of each article was then thoroughly reviewed.

### Quality assessment

The quality of methods was evaluated by S.H.Z.A. of the included articles using the 'Modified' Consolidated Standards of Reporting Trials (CONSORT) criteria of items for reporting in-vitro experiments, with minor adjustments suited to the study. The main domains are listed as follows: (1) structured summary in the abstract, (2) specific objectives or hypothesis, (3) study population, (4) further description of interventions, (5) primary and secondary outcomes, (6) results, (7) limitations, (8) sources of funding and (9) availability of protocol. Domains related to animal primary cultures, cell lines, and clinical trials were removed, however, domains on human stem cells were retained. Researchers' disagreements were addressed through discussion. The following markings were applied to each criterion: present (Yes) and not specified (No).

### Intended audience

This systematic review is aimed at researchers investigating human stem cell differentiation into osteoblasts and osteoclasts, especially those who employ in-vitro models. The information will be beneficial to scientists with a fundamental knowledge of cell biology

**Table 1  Combination of keywords for search string used in electronic databases in PubMed, WOS and Scopus.**

| No. | Keywords |
|---|---|
| 1 | Morphology analysis AND Osteoblast AND Osteoclast AND Human AND *In-vitro* |

and an interest in bone regeneration or disease modeling using stem cells. The study includes an overview of the most prevalent morphological techniques used in this field, making it a useful resource for both established researchers and those new to the field.

# RESULTS

## Data extraction results

Based on the keywords stated in Table 1, a search of the online databases PubMed, Web of Science (WOS), and Scopus yielded 71, six, and 178 articles, respectively. Based on the inclusion and exclusion criteria listed in Table 2, each prospective publication was evaluated separately. There were 248 articles left after six duplicates from the three databases were eliminated. A study procedure and 26 review articles made up the remaining 27 articles that were eliminated. A total of 207 articles were removed further; they included 77 irrelevant articles, 50 articles that featured animal research and did not fit the parameters of human study, and 80 studies that employed in-vivo methodologies. Fourteen publications in all met the requirements for qualitative synthesis in this systematic review after extensive screening. Figure 1 illustrates a flowchart of the article selection procedure.

## Study design

A total of 14 articles related to the morphological approach, published between 2013 and 2023 were selected. The human cultured stem cells used for the in-vitro studies include both peripheral blood mononuclear cells (PBMSC) (*Kleinhans et al., 2015*; *Maria et al., 2018*; *Steller et al., 2020*; *Grünherz et al., 2020*; *Zainol Abidin et al., 2023*) and mesenchymal stem cells (MSc) involved 5 studies (*Maria et al., 2017*; *Hashimoto et al., 2018*; *Liu et al., 2018*; *Ganguly et al., 2020*; *Nugraha et al., 2023*) whereas periodontal ligament stem cells (PDLSC) (*Di Vito et al., 2020*; *Frasheri et al., 2023*) and dental pulp stem cells (DPSC) (*Escobar et al., 2020*; *Escobar et al., 2023*) involved two studies (Table 3). The outcomes of the risk of bias assessment in Table 4 reflect the quality assessment analysis. The selected studies featured an abstract with a concise justification and a well-defined goal or hypothesis. The research also included details on the population, outcomes, and protocol or methodological description. Nine research lacked funding information, while seven studies did not disclose their limitations.

## Morphological results

Overall, this review demonstrates a variety of staining techniques that may be applied in the study of osteoblast and osteoclast development. The procedures used in the development of osteoblasts and osteoclasts in an in-vitro investigation using human stem cells are listed in Table 3 of the selected 14 studies. A total of seven different techniques were carried out
**Table 2  List of inclusion and exclusion criteria.**

| Article inclusion criteria | Article exclusion criteria |
|---|---|
| • English language articles. | • Animal study. |
| • Open access. | • Review articles. |
| • Stem cell research. | • Articles in other languages. |
| • Human cell study. | • Research Protocols. |
| • Osteoblast and osteoclast study. | • Nonrelated articles. |
| • *In-vitro* experimental design. | • *In-vivo* experimental design. |
| • Morphological analysis of cell differentiation. | • Non-stem cell research. |
| • Articles published between Year 2013–2023. | • Inappropriate methodologies. |
| • Appropriate methodologies. | • Book chapters. |
| • Original research articles. | • Redundant articles. |

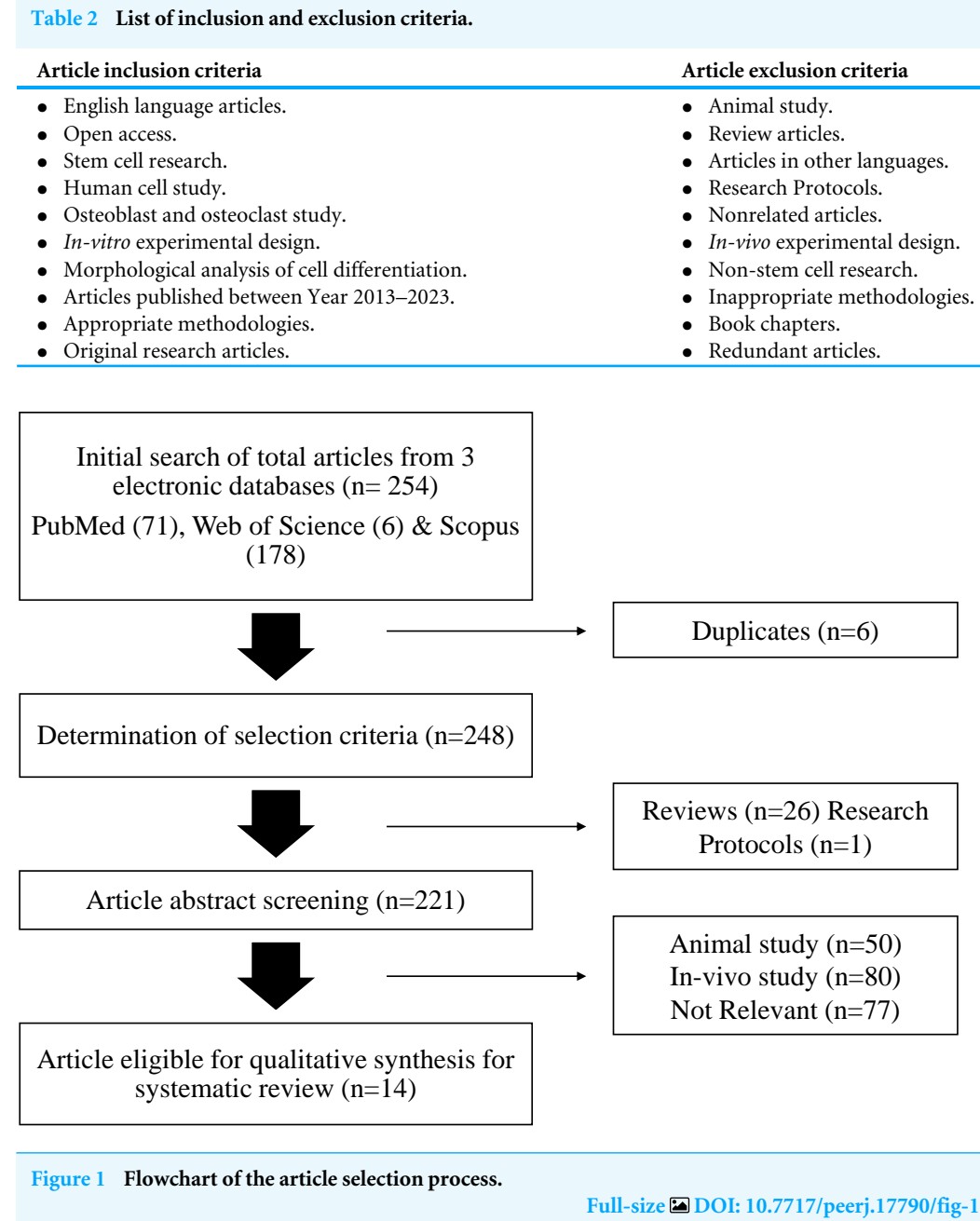

**Figure 1  Flowchart of the article selection process.**

and were repeatedly used (Fig. 2). The staining techniques were Alizarin Red S, von Kossa, alkaline phosphatase, immunofluorescence, giemsa and immunochemistry for osteoblast analysis, whereas TRAP, DAPI, immunohistochemistry and immunofluorescence for osteoclast activity.

Seven studies focus solely on the staining of osteoblast differentiation (*Escobar et al., 2020*; *Escobar et al., 2023*; *Ganguly et al., 2020*; *Di Vito et al., 2020*; *Frasheri et al., 2023*; *Zainol Abidin et al., 2023*; *Nugraha et al., 2023*), three studies only on morphological evaluation methods for osteoclast differentiation (*Kleinhans et al., 2015*; *Steller et al., 2020*;

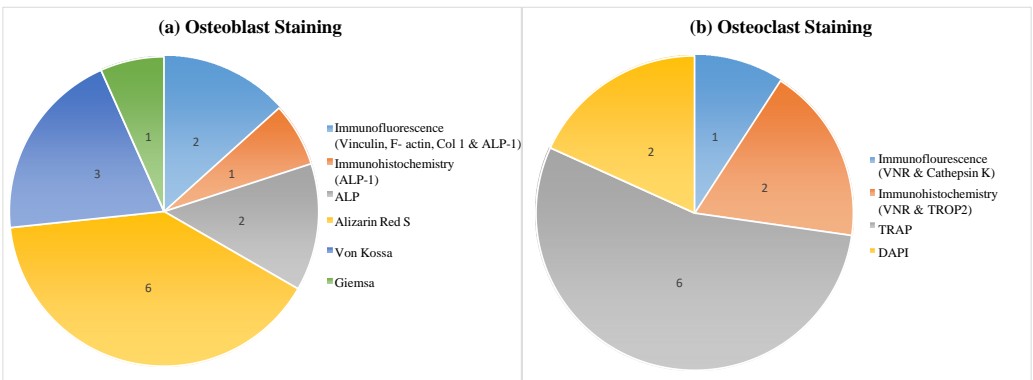

**Figure 2** **Variability in osteoblast and osteoclast morphology analysis across human stem cell sources: a staining comparison.** This figure marked the variable morphology analysis of osteoblasts (A) and osteoclasts (B) derived from different human stem cell sources, as visualized by various staining techniques. Alizarin Red S staining, with the highest citation count (six), effectively identifies osteoblasts, while TRAP staining (also six citations) is the most used and cited method for osteoclast detection.

*Grünherz et al., 2020*) and another four articles involved qualification analysis of both osteoblast and osteoclast differentiation (*Hashimoto et al., 2018*; *Maria et al., 2018*; *Liu et al., 2018*) (Table 3). All the staining techniques were proven to be successful in evaluating the presence of osteoblast and osteoclast differentiation in cells.

## Alizarin Red S staining

Alizarin Red S (ARS) staining emerged as the dominant method for analyzing osteoblast differentiation in studies utilizing various stem cell types, including DPSCs, PDLSCs, PBMSCs, and MSCs (six articles). This technique offers a reliable and straightforward approach to visualize mineralized nodules, a characteristic of mature osteoblasts, through the detection of calcium deposits-stained deep red by ARS. Alizarin Red S is a chelating dye with an affinity for calcium. It will bind to the calcium deposits that are associated with mineralization in the bone matrix formed by osteoblast (*Wang et al., 2006*). The presence and intensity of this type of staining will indicate the extent of mineralization, thus reflecting osteoblast activity. Alizarin Red S offers a specific, reliable technique for assessing osteoblast activity through bone matrix mineralization.

Several studies proved that ARS staining is a useful tool to evaluate osteogenic differentiation induced by various compounds. Effective osteoblast maturation was indicated by increased calcium deposition in human MSCs under melatonin treatment in osteogenic medium (OS+) or MSDK (melatonin, strontium, vitamin D3, and vitamin K2) (*Maria et al., 2017*; *Maria et al., 2018*). Comparably, PDLSCs treated with zoledronic acid (ZOL) had more ARS-positive nodules, indicating improved mineralization (*Di Vito et al., 2020*). The osteoblast development that Vitamin D and E caused in DPSCs was confirmed by *Escobar et al. (2020)* using ARS staining. This was demonstrated by changes in morphology as well as the creation of calcified nodules. Further demonstrating the value of this method, relaxin administration also enhanced osteoblastic morphology in DPSCs as seen by ARS (*Escobar et al., 2023*). Furthermore, *Nugraha et al. (2023)* qualitatively used

**Table 3  Morphological staining approaches on human stem cells.**

| Cell type | Article | Osteoblast staining | Osteoclast staining |
|---|---|---|---|
| Peripheral Blood Monocyte Stem Cells (PBMSC) | *Kleinhans et al. (2015)* | | • TRAP<br>• Immunohisto-chemistry (VNR)<br>• Immunofluorescence (VNR & Cathepsin K) |
| | *Grünherz et al. (2020)* | | • TRAP<br>• Immunohistochemistry (TROP2) |
| | *Zainol Abidin et al. (2023)* | Von Kossa | |
| | *Maria et al. (2018)* | Alizarin Red S | • TRAP |
| | *Steller et al. (2020)* | | • TRAP<br>• DAPI |
| Mesenchymal Stem Cell-s/Mesenchymal Stem Cell line (MSC) | *Liu et al. (2018)* | • Immunohistochemistry (*Col 1*) | • DAPI |
| | *Nugraha et al. (2023)* | • Alizarin Red S | |
| | *Ganguly et al. (2020)* | • ALP<br>• Giemsa<br>• Immunofluorescence (Vinculin & F-actin) | |
| | *Maria et al. (2017)* | • Alizarin Red S | • TRAP |
| | *Hashimoto et al. (2018)* | • Von Kossa | • TRAP |
| Periodontal Ligament Stem Cells (PDLSC) | *Frasheri et al. (2023)* | • Von Kossa<br>• Cell Morphology | |
| | *Di Vito et al. (2020)* | • Alizarin Red<br>• Immunofluorescence (*Col I*) | |
| Dental Pulp Stem Cells (DPSC) | *Escobar et al. (2020)* | • Alizarin Red S | |
| | *Escobar et al. (2023)* | • Alizarin Red S<br>• ALP | |

ARS to evaluate osteogenesis in MSCs obtained from the umbilical cord. It's interesting to consider that when compared to other bone replacements, freeze-dried bovine bone (FDBB) supplementation produced the highest calcium deposition, suggesting that it may be beneficial for promoting osteoblast activity (*Nugraha et al., 2023*). In human stem cell research, ARS staining showed to be an effective and sensitive approach for analyzing osteoblast development. Its capacity to directly observe mineralized deposits sheds light on various differentiation techniques for future advancements in bone regeneration therapy.

## Von Kossa staining

Von Kossa technique utilizes silver nitrate to precipitate silver phosphate at sites containing calcium phosphate. While targeting mineralization, it can also react with other metals, leading to potential background staining (*Lee et al., 2017*). This type of staining will provide a visual of mineralized nodules, although lacks the specificity of ARS for bone-related mineralization. Therefore, von Kossa staining can be a preliminary indicator of mineralization and requires confirmation with other approaches due to potential non-specificity. The von Kossa staining method, although less commonly utilized in

**Table 4  Risk of bias assessment using CONSORT.**

| Domain<br>Author (year) | 1 | 2 | 3 | 4 | 5 | 6 | 7 | 8 | 9 |
|---|---|---|---|---|---|---|---|---|---|
| *Kleinhans et al. (2015)* | Yes | Yes | Yes | Yes | Yes | Yes | No | No | Yes |
| *Grünherz et al. (2020)* | Yes | Yes | Yes | Yes | Yes | Yes | Yes | No | Yes |
| *Zainol Abidin et al. (2023)* | Yes | Yes | Yes | Yes | Yes | Yes | Yes | Yes | Yes |
| *Maria et al. (2018)* | Yes | Yes | Yes | Yes | Yes | Yes | No | No | Yes |
| *Steller et al. (2020)* | Yes | Yes | Yes | Yes | Yes | Yes | No | No | Yes |
| *Liu et al. (2018)* | Yes | Yes | Yes | Yes | Yes | Yes | Yes | Yes | Yes |
| *Nugraha et al. (2023)* | Yes | Yes | Yes | Yes | Yes | Yes | No | No | Yes |
| *Ganguly et al. (2020)* | Yes | Yes | Yes | Yes | Yes | Yes | Yes | No | Yes |
| *Maria et al. (2017)* | Yes | Yes | Yes | Yes | Yes | Yes | Yes | Yes | Yes |
| *Hashimoto et al. (2018)* | Yes | Yes | Yes | Yes | Yes | Yes | No | No | Yes |
| *Frasheri et al. (2023)* | Yes | Yes | Yes | Yes | Yes | Yes | No | No | Yes |
| *Di Vito et al. (2020)* | Yes | Yes | Yes | Yes | Yes | Yes | Yes | Yes | Yes |
| *Escobar et al. (2020)* | Yes | Yes | Yes | Yes | Yes | Yes | No | No | Yes |
| *Escobar et al. (2023)* | Yes | Yes | Yes | Yes | Yes | Yes | Yes | Yes | Yes |

osteoblast differentiation, was identified in three articles (*Hashimoto et al., 2018*; *Frasheri et al., 2023*; *Zainol Abidin et al., 2023*) as detailed in Table 3 and Fig. 2. Assessment of calcium mineralization rates in PBMSCs following treatment with various concentrations of *Piper sarmentosum* ethanolic extract was conducted using the von Kossa technique. The findings demonstrated that the concentration of 50 µg/mL exhibited the most significant induction of osteoblast differentiation. This extract outperformed the positive control group, which involved cells treated with a combination of 50 µg/mL ascorbic acid and 10 mM $\beta$-glycerophosphate (*Zainol Abidin et al., 2023*), in inducing osteoblastic differentiation.

Additionally, the *in-vitro* study focusing on PDLSCs demonstrates the regulatory role of hyaluronic acid (HA) in mineralization using the von Kossa staining technique. Further investigation suggested that among various molecular weights tested, low molecular weight hyaluronan (15–40 kDa) exhibited the highest calcium mineralization, surpassing high molecular weight hyaluronan (>950 kDa), medium molecular weight hyaluronan (75–350 kDa), and ultralow molecular weight hyaluronan (4–8 kDa) (*Frasheri et al., 2023*). Moreover, in a separate study was found that micro-RNA-940 (hsa-miR-940) expressed by prostate cancer cells stimulated osteogenic differentiation of human MSCs in-vitro and induced extensive osteoblastic lesions within the bone metastatic microenvironment *in-vivo* (*Hashimoto et al., 2018*).

## Alkaline phosphatase (ALP) staining

ALP is an enzyme marker for early osteoblast differentiation. It plays a role in bone mineralization by hydrolysing the phosphate group. The increment of ALP activity through enzyme assay or antibody-based approach suggested the presence of pre-osteoblasts or immature osteoblasts actively producing the enzymes. However, ALP is also present in other cell types such as in the cytosol of liver cells and the canalicular membrane of hepatocytes

(*Green & Sambrook, 2020*), which makes ALP not the definitive marker for osteoblast. Therefore, required confirmation with other techniques due to potential non-specificity.

ALP staining has also been used to examine stem cell differentiation towards osteoblasts by measuring qualitatively the expression levels of ALP through staining (*Ganguly et al., 2020*; *Escobar et al., 2023*). One article each resulted in high levels of ALP-stained cells representing successful osteoblast differentiation of dental tissues, *i.e.,* DPSC and MSC that led to early calcification during bone development (*Ganguly et al., 2020*; *Escobar et al., 2023*).

## Giemsa staining

Giemsa staining is versatile and extensively used during cell biology study that provides useful insights into cellular morphology, particularly in osteoblasts. In principle, Giemsa staining is a general-purpose stain that differentiates various cellular components based on their acidity (stained by Methylene Blue) or basicity (stained by Eosin Y). It highlights cellular components such as nuclei, cytoplasm and some extracellular matrix components (*Bancroft & Gamble, 2013*). This type of staining provides general cell morphology but does not specifically identify osteoblasts, hence, only be used to visualize osteoblast morphology alongside other techniques of osteoblast analysis. This staining method allows for the imaging of cell morphology and the tracking of changes in osteoblast morphology during stem cell differentiation such as human mesenchymal stem cells osteoinductive analysis of the insect-derived protein isolate from *Protaetia brevitarsis seulensis* (*Ganguly et al., 2020*).

## Immunohistochemistry or immunofluorescence analysis

The investigation of human stem cell differentiation into osteoblasts and osteoclasts was greatly aided by immunochemistry techniques. Advances in regenerative medicine and bone disease therapeutics are made possible by these approaches, which provide useful knowledge about the complex processes of bone growth and resorption. Currently in use are two distinct immunochemistry methods: immunofluorescence (IF) and immunohistochemistry (IHC). The IF technique uses fluorescently labelled antibodies to visualize protein expression within individual cells. Its high sensitivity and ability to show cellular localization made it ideal for analyzing osteoblast and osteoclast differentiation in cell cultures (*Kleinhans et al., 2015*; *Ganguly et al., 2020*; *Di Vito et al., 2020*). For example, Vinculin and F-actin antibodies were used to assess cytoskeletal changes during MSC differentiation into osteoblasts (*Ganguly et al., 2020*), while *Di Vito et al. (2020)* employed immunofluorescence staining for Collagen I and ALP to track osteoblast differentiation in PDLSCs.

On the other hand, the immunohistochemistry technique uses enzyme-linked antibodies and chromogenic substrates to visualize protein expression in tissue sections. Its suitability for tissue analysis made it valuable for studying osteoclast differentiation in PBMSCs (*Kleinhans et al., 2015*; *Grünherz et al., 2020*). For instance, immunohistochemistry analysis using VNR (vitronectin receptor $\alpha V \beta 3$) antibodies was employed to identify osteoclasts in these cells (*Kleinhans et al., 2015*). Additionally, TROP 2 antibodies were used in immunohistochemistry to mark osteoclasts in PBMSC-derived bone tissues (*Grünherz*

*et al., 2020*). The expression patterns of protein markers associated with the differentiation of osteoblasts and osteoclasts using immunochemistry will expand our understanding of the processes involved in bone remodeling and may help create new approaches to treating bone diseases and using regenerative medicine.

### TRAP staining

TRAP is a key enzyme mature osteoclast produces by hydrolyzing the bone matrix's phosphate ester (*Boyde & Jones, 1992*). TRAP staining technique will directly visualize the enzyme activity, providing the osteoclast functional activity. The advantage of using TRAP is that the enzyme directly identifies osteoclast bone resorption activity and is relatively easy to perform and affordable (cost-effective) compared to antibody-based approaches.

It was discovered that six studies used TRAP labelling techniques for osteoclast cells. The levels of TRAP expression were assessed since this protein is a recognized indicator of terminally differentiated osteoclasts and the activity of bone resorption they exhibit. TRAP staining approaches were also applied to PBMSCs (*Kleinhans et al., 2015*; *Maria et al., 2018*; *Grünherz et al., 2020*) and MSC (*Maria et al., 2017*; *Hashimoto et al., 2018*; *Steller et al., 2020*) which showed successful differentiation into osteoclasts.

### DAPI staining

DAPI (4′,6-diamidino-2-phenylindole) staining was a simple and widely used technique for visualizing and quantifying nuclei in cell cultures. DAPI is a fluorescent stain that strongly binds to the double-stranded DNA in the cell nucleus (*Chazotte, 2011*). This staining allows visualization of all cell nuclei present in the sample although able to determine the number of nuclei, however unable to differentiate between osteoclast with other types of cells that are involved during bone formation.

During the osteoblast differentiation process from stem cells, DAPI staining provides valuable insights into cell number, morphology, and distribution. Assessing this process accurately requires reliable and reproducible methods. DAPI staining fulfils this role by enabling cell counting which DAPI binds to DNA in the nucleus, allowing quantification of total cell number and changes during differentiation. DAPI staining also reveals changes in nuclear size and shape, hence enabling nuclear morphology analysis which can indicate differentiation stage and cellular stress. The existence of osteoclast could be observed using DAPI in both monoculture and indirect co-culture of osteoblast and fibroblast cells with PBMSC (*Steller et al., 2020*). Finally, DAPI staining also provides cell distribution assessment by visualizing the representation of cell distribution and potential clustering patterns associated with differentiation in the establishment and validation of the in-vitro co-culture model (*Liu et al., 2018*; *Steller et al., 2020*).

## DISCUSSION

The cultured cells used for the in-vitro studies include PBMSC, MSC, PDLSC and DPSC. The most common cells used were the human PBMSC followed by the MSC (five studies) and another two studies each involving PDLSCs and DPSC.

PBMSC is the adult stem cell obtained from peripheral blood hematopoietic stem cells (HSC). The most well-studied tissue-specific stem cells with promise for regenerative

therapy were hematopoietic stem cells (*Zakrzewski et al., 2019*). Osteoclast precursor cells have been employed in in-vitro investigations using monocytes generated from HSC, which make up about 10–20% of peripheral blood (*Ansari, Ito & Hofmann, 2021*). Furthermore, it was demonstrated that HSC and monocytes may be isolated and purified according to the expression of specific surface markers including CD34 and CD14. But unlike MSC, the processes for isolating cells take a long time. This might result in a limited quantity of separated cells, requiring the need for more aspirated bone marrow or peripheral blood (*Ansari, Ito & Hofmann, 2021*).

The presence of osteoblast and osteoclast cells will involve specific staining procedures such as ARS, von Kossa staining, ALP, immunofluorescence (vinculin, F-actin and *Col1* antibodies) and Giemsa staining for osteoblast, on the other hand, TRAP, immunofluorescence, immunochemistry and DAPI staining for the detection of osteoclasts.

## Osteoblastic differentiation detection

The most frequently used staining technique is ARS which was found in six articles. It was mainly used to assess osteoblast differentiation through mineralization staining of calcium deposits. The Alizarin will bind to the osteoblast calcium ions and stain them in red. The red colour which was achieved during staining indicates successful calcium deposition and the absence of staining indicates negative results. Cells show positive reactions with ARS indicating the expression of calcium-bound bone morphogenic proteins such as osteonectin. Therefore, ARS can be classified as an osteoblast detection standard that has been traditionally used during in-vitro studies of osteoblasts for analyzing the quantification of mineralization. However, this method presents limitations, such as the need to fix the cells after staining which only allows one sample to be analyzed once (*Serguienko, Wang & Myklebost, 2018*). ARS also makes early differentiation hard to detect due to its moderate sensitivity to calcium ions. In addition, the removal of background signals from nonspecific dye-binding will make the detection of the early stage of stem cell osteoblastic differentiation a challenging process. Concentrations of ARS reflecting osteoblast mineralization activity can be determined by measuring the absorbance which has to be normalized and compared between control groups. This method increases variability and time-consuming process (*Maria et al., 2018*). Despite the limitations, ARS staining had been the most widely used approach for osteoblastic differentiation detection due to its high sensitivity towards calcium deposits, which enables ARS to provide more reliable results compared to other approaches that stain weakly.

Von Kossa staining meanwhile was used to measure the mineralization of the calcium matrix secreted by osteoblast. Von Kossa staining is broadly used in histological analysis to detect cellular calcium deposits (*Blair et al., 2016*). The underlying principle of this coloration is based on the transformation of calcium salts into silver salts; silver ions, reinforced by silver nitrate solution, would replace calcium ions by their binding with phosphates. Once there is sufficient light to generate different intensities of the brown or black color, metallic silver deposits are formed as a result of the photochemical breakdown of the silver phosphates (*Meloan & Puchtler, 1985*). The presence of mineralized tissues that encircle osteoblast-like cells, or osteoids, and make up the calcium bone matrix may

be found by von Kossa staining. The homogeneous dark or black staining of the calcified cartilage and mineralized bone is the disadvantage of the von Kossa stain. Determining the structural features inside mineralized bone matrix tissue, such as remodeling units or calcified cartilage remains, using von Kossa staining was, therefore, a difficult process.

Von Kossa staining can indicate the presence of minerals that resemble bone, but it might not be sufficient to identify and quantify these structures conclusively, especially when it comes to features like calcified cartilage remains or remodeling units. Additional methods such as electron microscopy (EM) or Fourier transform infrared spectroscopy (FTIR) or more specific calcium bone matrix staining, can provide complementary information to confirm the existence and quality of calcium phosphate in the bone (*Bonewald et al., 2003*). Calcium deposited area demonstrated by staining with von Kossa of the differentiated cells could also be quantified and statistically analyzed by using ImageJ software; an automated image analysis program used for computerized morphology analysis of cells (*Malhan et al., 2018*; *Zainol Abidin et al., 2023*).

ALP staining techniques have been found in two studies throughout this systematic observation. The (5-Bromo-4-chloro-3-indolyl phosphate/Nitro blue tetrazolium) (BCIP/NBT) blue stain was also used for the assessment of ALP activity. To quantify ALP activity, both the BCIP/NBT liquid substrate and alkaline phosphatase-yellow liquid substrate system were used (*Chiarella et al., 2018*). It acts as a substrate to reveal alkaline phosphatase in immunohistochemical analysis as well as helps in detecting specific immune complexes. The BCIP which was the substrate for ALP will be oxidized by NBT after dephosphorylation and generates a dark blue stain. The NBT will then be reduced to form a dark blue precipitating stain which functions to make the colour of the reaction more intense, hence the detection becomes easier. The staining result was considered positive if it showed precipitates of reduced products in dark blue generated by ALP. ALP is a membrane-bound metalloenzyme that catalyzes the hydrolysis of phosphomonoesters in an alkaline environment (*Sharma, Pal & Prasad, 2014*). ALP is an early osteogenic marker that is used to examine cellular differentiation towards osteoblasts.

Active osteoblasts will show an increase in ALP activity, during in-vitro bone matrices formation. ALP activity can easily be measured using 5-Bromo-chloro-3-indolylphophate/Nitro Blue Tetrazolium (BCIP/NBT) as a substrate and generate a product that will stain cells as dark blue-violet. Active ALP will cleave BCIP to release a colourless intermediate followed by a reaction with NBT to form a dark-blue violet precipitate. However, this fixation approach will lead to irreversible inactivation of ALP hence, the active enzyme will be difficult to recover when using this technique. ALP negative results will show colourless or faintly bluish, whereas ALP positive results are indicated by dark blue violet. The higher the ALP enzyme activity, the more intense the colour will be generated. This type of staining is quite sensitive to fixation and is denatured by moderately high temperatures and will lose its colour after some time, thus storage of ALP-stained cells for more than a year is not recommended. A further complication is that the decalcification procedures remove the magnesium and zinc ions necessary to reduce ALP enzyme sensitivity. Thus, pre-incubation with magnesium chloride can also restore the enzyme activity in decalcified tissue samples (*Miao & Scutt, 2002*). Additionally,

azo-dye techniques can prevent the diffusion of reaction products and possible false localization brought on by the prevalence of phosphate groups in tissue components. Immunofluorescence was also being utilised during osteoblast differentiation from MSC and PDLSC. In the MSC differentiation approach, the existence of osteoblast cells was observed using Vinculin and F-actin antibodies whereas for PDLSC, *Col1* antibody was utilized to observe osteoblast activity. Giemsa staining has only been used once together with ALP and Immunofluorescence approaches to analyse osteoblast differentiation from MSC (*Ganguly et al., 2020*).

ARS will specifically stain calcium deposits during bone matrix synthesized by osteoblast to produce an accurate osteoblast activity compared to others visually approached such as von Kossa, ALP, Giemsa, immunohistochemistry, or immunofluorescence. The specificity of bone nodule staining was reduced by von Kossa staining as this procedure identified osteoblast mineralization and reacted with other metals, producing a background. Since ALP is an enzyme marker exclusive to osteoblasts and is found in other cell types, it does not always indicate osteoblast activity. Giemsa staining does not effectively distinguish osteoblasts from other cell types and is less selective for osteoblasts. On the other hand, although immunohistochemistry and immunofluorescence require specific and accurate antibodies, they are more expensive, tedious and time-consuming and require well-trained experts to interpret accurately. Immunological analysis of osteoblast involved four biomarkers, *i.e.,* Vinculin, F-actin, Col-1 and ALP. ALP is the most commonly used marker for osteoblast assessment using an immunological assay due to the enzyme specificity for the osteoblastic lineage particularly during an early stage of differentiation (*Feldman et al., 1993*). Vinculin, F-actin and Col-1 were markers with broader expression across various cell types and not specifically present in osteoblast (*Ziegler, Liddington & Critchley, 2006*; *Rao et al., 1990*; *Henriksen & Karsdal, 2016*) (Table 5).

## Osteoclastic differentiation detection

The biological marker known as TRAP was used to identify differentiated osteoclasts and the activity of bone resorption they exhibit. Osteoclast activity was evaluated by TRAP enzyme histochemistry (*Sheehan & Hrapchak, 1980*). This stain's principle involves using naphthol AS phosphates in conjunction with fast garnet GBC salts (diazonium dye) to detect acid phosphatase (*Newa et al., 2011*). Measuring osteoclastic differentiation and TRAP-releasing activity is the primary purpose of TRAP staining. The quantity of TRAP deposited by osteoclasts is correlated with their differentiation, as seen by the purple staining. The TRAP negatively unstained cells in differentiated culture signify the absence of osteoclasts. An advantage of TRAP staining is that its histochemical staining outcome can be easily detected using light microscopy as it results in a purple colour precipitate that is suitable for visualization as compared with unstained cells. While the conventional TRAP stain is simple to apply, it has a drawback in that it is challenging to combine with other particular stains, including those employed in immunohistology techniques. However, a fluorescence-based TRAP staining approach could be combined with other fluorescence-based stains such as an actin stain using fluorescence-labelled phalloidin (*Filgueira, 2004*). TRAP stain for PBMCs could also be substituted with an enzyme-labeled fluorescent 97 (ELF97), a

**Table 5  Morphological assay and staining approaches for osteoblast and osteoclast differentiation in human stem cell: strength and weakness.**

| Staining techniques | Differentiation analysis | Strengths | Weaknesses |
|---|---|---|---|
| Alizarin Red S | Calcium Mineralization (Osteoblast) | ● Simple and specific in detecting calcium deposits<br>● Qualification and quantification of mineralization | ● Unable to differentiate early stage of osteoblast |
| Von Kossa | Calcium Mineralization (Osteoblast) | ● Existence of Calcium matrix | ● Less sensitive and unable to differentiate different types of minerals |
| Alkaline Phosphatase (ALP) | Early stage of Osteoblast | ● Detect ALP enzyme production at the early stage of osteoblast differentiation | ● Not specific to osteoblast |
| Giemsa | Cell morphology (Osteoblast). | ● Basic information on cell shape and size | ● Not specific to osteoblast |
| Immunological | Biomarkers:<br>● Osteoblast (Vinculin, F-actin, Col-1 & ALP)<br>● Osteoclast (VNR, Cathepsin K, TROP 1) | ● Highly specific on the individual markers<br>● Common markers for Osteoblast (ALP) and Osteoclast (Cathepsin K) | ● Needs experts for experiments and interpretations<br>● Expensive |
| TRAP | Activity and function (Osteoclast) | ● Detect mature osteoclast | ● Unable to differentiate between active and inactive osteoclast |
| DAPI | Nucleus Morphology (Osteoclast) | ● Simple<br>● Effective qualification and quantification analysis | ● Limited information on osteoclasts' activity and function |

fluorescent method to detect TRAP-positive granules (*Borciani et al., 2022*). TRAP staining has now evolved as the only current histological approach in the detection of osteoclastic differentiation because of its unique ability to specifically target high expressions of TRAP cytochemical marker directly in osteoclast cells. The other alternative morphological staining approaches for the investigation of osteoclast activity are immunohistochemistry and immunofluorescence techniques. Two articles utilized the immunohistochemistry approach, both employing PBMSC while utilizing different antibodies. Specifically, one study utilized antibodies against VNR to assess osteoclastogenesis and resorption activity in PBMSC cultures on both polystyrene and natural extracellular bone matrix, in both 2D and 3D settings (*Kleinhans et al., 2015*). Conversely, another study employed the TROP2 antibody to observe PBMSC differentiation into osteoclasts on mineralized surfaces and osteomalacic bone obtained from patients with osteomalacia. Osteomalacia is a condition characterized by a defect in bone mineralization, resulting in alterations to the bone surface characteristics, leading to the development of soft and weak bone tissue. These diverse approaches of osteoblast and osteoclast morphological analyses contribute valuable nuances to our understanding of osteoblast and osteoclast activity in varying cellular and matrix contexts, providing a more comprehensive perspective on potential applications in the field of osteogenesis-related research and therapeutic interventions.

TRAP staining offers a specific, cost-effective, straightforward approach to evaluate osteoclast activity through direct correlation with their bone resorption function. DAPI

staining technique will stain the cell nuclei which does not distinguish between osteoclasts and other cell types during bone remodelling. DAPI can reveal the number of nuclei present but cannot specifically indicate the cell belongs to osteoblast. On the other hand, the immunological methods (immunohistochemistry and immunofluorescence) will only use a more in-depth examination of the signalling pathways or osteoblast subpopulation. The immunological analysis of osteoclast involved three markers, *i.e.,* VNR, cathepsin K and TROP 2. Among these markers, cathepsin K appears to be the most commonly used in osteoclast analysis. Cathepsin K is an enzyme that plays an important role during osteoclast-mediated bone resorption as the enzyme is specifically expressed in active osteoclasts (*Gruber, 2015*; *Shahrul Hisham et al., 2010*). WNR and TROP 2 are less specific markers as these markers are also expressed in other cells such as endothelial and epithelial cells, respectively. Although, immunological techniques, *i.e.,* immunohistochemistry and immunofluorescence provide more detailed information on specific osteoclast markers; TRAP staining remain the preferred choice for initial observation and routine osteoclast identification due to its simplicity and focus on a key functional aspect (Table 5).

## CONCLUSION

This systematic review provides useful insights for future researchers studying the differentiation of human stem cells, with an emphasis on osteoblast and osteoclast staining techniques. The systematic analysis includes six types of osteoblast staining and four different osteoclast morphological techniques. The Alizarin Red S staining technique is the most common and widely used method for determining osteoblast differentiation, followed by von Kossa, ALP, and immunofluorescence. In terms of osteoclast differentiation detection; TRAP staining is the most often used for morphological analysis, followed by immunohistochemistry. This review can provide a significant resource for giving insight into the current cell morphological and tissue histology techniques.

## ACKNOWLEDGEMENTS

We are grateful to our undergraduates Ameedtri and Leesha, as well as Zahidah, a postgraduate Ph.D. student in the Genetics program, for their significant contributions to this analysis. They gave critical input on the text and assisted with data analysis. We value their intellectual input and efforts, which significantly improved the quality of this article.

### Funding

This work was supported by grant GP-K007744 from the Universiti Kebangsaan Malaysia, CRGS/URGS/2023_005 from University of Cyberjaya and FRGS/1/2018/STG05/CUCMS/02/1 from the Ministry of Higher Education, Malaysia. The funders had no role in study design, data collection and analysis, decision to publish, or preparation of the manuscript.

## Grant Disclosures

The following grant information was disclosed by the authors:
The Universiti Kebangsaan Malaysia: GP-K007744.
University of Cyberjaya: CRGS/URGS/2023_005.
Ministry of Higher Education, Malaysia: FRGS/1/2018/STG05/CUCMS/02/1.

## Competing Interests

The authors declare there are no competing interests.

## Author Contributions

- Shahrul Hisham Zainal Ariffin conceived and designed the experiments, performed the experiments, analyzed the data, prepared figures and/or tables, authored or reviewed drafts of the article, and approved the final draft.
- Rohaya Megat Abdul Wahab performed the experiments, analyzed the data, authored or reviewed drafts of the article, and approved the final draft.
- Muhammad Abdul Razak analyzed the data, authored or reviewed drafts of the article, and approved the final draft.
- Muhammad Dain Yazid analyzed the data, authored or reviewed drafts of the article, and approved the final draft.
- Muhammad Ashraf Shahidan analyzed the data, authored or reviewed drafts of the article, and approved the final draft.
- Azizi Miskon analyzed the data, authored or reviewed drafts of the article, and approved the final draft.
- Intan Zarina Zainol Abidin conceived and designed the experiments, performed the experiments, analyzed the data, prepared figures and/or tables, authored or reviewed drafts of the article, and approved the final draft.

## Data Availability

This is a systematic review/meta-analysis.

## Supplemental Information

Supplemental information for this article can be found online at http://dx.doi.org/10.7717/peerj.17790#supplemental-information.

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
