# Peer review of "Evaluation of in vitro osteoblast and osteoclast differentiation from stem cell: a systematic review of morphological assays and staining techniques"

_PeerJ, doi:10.7717/peerj.17790_

## Round 0.1 · original submission · Minor Revisions

· Academic Editor

Minor Revisions

The manuscript provides a summary of the recently published work for evaluating osteoblasts and osteoclasts differentiation and is relevant to PeerJ's readership. There are some minor concerns that the reviewers have raised which would improve overall quality of the manuscript and needs to be addressed.

Reviewer 1 ·

Basic reporting

- The authors have included sufficient references, background and context in this article. The only comment I have here is that the title may need to be changed from 'morphological techniques' to something that reflects the nature of assays/stains used. This is because morphological techniques generally refer to image processing operations in histopathology and may introduce ambiguity.
- More background on WHY certain stains are used more than others needs to be included.

Experimental design

- The authors have conducted a thorough review of the staining techniques and have done a great job at including some very important references.
- The research question has been defined well as per PRISMA guidelines. While the authors are looking to analyze trends in commonly used staining techniques, some of the articles published previously have mentioned Alizarin S and TRAP staining as the gold standards. Eg. (https://www.ncbi.nlm.nih.gov/pmc/articles/PMC9821450/)
- The methods described here only aim to analyze the trends in staining techniques, and not standardization as the authors suggest at the beginning of the article.
- Since review articles provide a summary of current trends/update techniques, it is unclear what is the added value of this article when compared to previously published reviews.
- Since different staining techniques are better-suited to answer different questions, eg. rate/extent of mineralization vs enzymatic activity, analysis of trends does not fill a knowledge gap.

Validity of the findings

No comment

Additional comments

No comment.

·

Basic reporting

Ariffin et al have used a systematic approach to comprehend the existing and the most prevalent morphological techniques to evaluate osteoblasts and osteoclast differentiation. It is a great advantage to keep the focus of the review on the differentiation methods studied in the last decade using human stem cells. The Introduction looks adept and the Background is explained in detail. The Introduction is followed by coherently articulating the posed research question. The text is backed up by strong, clear and scientifically relevant thoughts.

Experimental design

The entire process of literature research is outlined well and is easy to understand for the readers.
The method has been thoroughly described with the selection criteria for the journals studied
The review highlights the work from 14 journal articles that are worth discussing in depth. Figures and Tables contribute towards the rigorous investigation that was put into the analysis.

Validity of the findings

The research question asked in the review article is addressed fully to the best of the abilities in the conclusion. This discussion looks well organized and describes the limitations of the most frequently used techniques and the alternate methods that could be used.

Additional comments

Some minor revisions that could be taken into consideration:
1. The acronyms in lines 185-188 for PDLCs, DPSCs, and MSCs have been introduced and are still repeated a few times in the study. The repetition can be noticed in lines 213-214 of the Results section and in lines 313-315 of the Discussion section. PBMSC acronym is again introduced in line 428. This seems redundant use of the text space. There is a discrepancy in the abbreviation used for Mesenchymal stem cells that can be corrected.
2. Similar repetition of Alizarin Red S (ARS) acronym that was first introduced in line 211. It is noticed again in lines 329 and 334 of the Discussion section.
3. Do authors mean von Kossa staining is the sole technique appropriate in Line 365? Other relevant techniques are mentioned. Probably the sentence can be rephrased to bring out the meaning intended.
4. Figure 1 mentions 71 articles from PubMed database in the flowchart whereas it says 70 articles in the text of line 172 in the data extraction results section of Results. Is there a reason for that?
5. In line 388, BCIP/NBT can be written without the parentheses. The sentence is pointing to the substrate BCIP/NBT.

Reviewer 3 ·

Basic reporting

The manuscript summarises the common morphological techniques for bone research. The current analysis is acceptable, however requires more elaboration on several aspects before I can recommend acceptance.

Experimental design

The data search, screening and analysis were performed accordingly.

Validity of the findings

The data included is current 2013-2023 and was analysed comprehensively.

Additional comments

- Title: I recommend including the phrase "in vitro" in the title, as the study exclusively focuses on in vitro research.
- It would be beneficial to include a brief description of the principle of staining in the results section to better explain the findings for each staining technique.
- In the results section, for immunohistochemistry (IHC) and immunofluorescence (IF), the authors can emphasize the most commonly used bone markers for staining.
- Including representative images for each staining technique, along with details of the observations, would be helpful for future bone researchers to better understand the review findings.

---

## Round 0.2 · accepted · Accept

· Academic Editor

Accept

The revised manuscript has incorporated all the reviewers' comments and has improved the manuscript significantly to meet the standards of the journal.

Reviewer 3 ·

Basic reporting

The author has highlighted all the concerns, and amended version of the manuscript is acceptable for publication.

Experimental design

Ok.

Validity of the findings

Ok.

Additional comments

NA